# Determination of the chemical equator from GEOS-Chem model simulation: a focus on the Tropical Western Pacific region

Xiaoyu Sun[1], Mathias Palm[1], Justus Notholt[1], Katrin Müller[2], and Jonas Hachmeister[1]

[1]Institute of Environmental Physics, University of Bremen, Otto-Hahn-Allee 1, 28359 Bremen, Germany
[2]Alfred Wegener Institute, Helmholtz Centre for Polar and Marine Research, Telegrafenberg A43, 14473 Potsdam, Germany

**Correspondence:** Xiaoyu Sun (xiaoyu_sun@iup.physik.uni-bremen.com)

**Abstract.** The Tropical Western Pacific (TWP) plays an important role for global stratosphere-troposphere exchange and is an active region of the interhemispheric transport (IHT). Common indicators for transport between the hemispheres like the tropical rain belt are too broad or lack precision in the TWP. In this paper, we provide a method to determine the atmospheric chemical equator (CE), which is a boundary for air mass transport between the two hemispheres in the tropics. This method used the model output from an artificial passive tracer simulated by the chemical transport model GEOS-Chem in the troposphere. We investigated the movement of the CE in the tropics, which indicates the migration of atmospheric circulation systems and air mass origins. Our results show the CE in different time scales suggesting that the different features of the IHT in different regions are highly related to the variation of the circulation systems. We compared the CE with the tropical wind fields, indicating that the region of IHT does not coincide with the convergence of the 10-m wind fields, in the tropical land sectors and the TWP region. We compared the CE with the atmospheric composition such as satellite data of $CH_4$ and model simulation of $SF_6$. The results show that the CE and north-south gradient of $CH_4$ in the Indian Ocean in January are well consistent with each other which indicates CE has a good potential to estimate the IHT inferred by observations. We discussed the vertical extent and the meridional extent of the IHT. We find that the vertical structure above 2 km have a slight northern tilt in the Northern Hemispheric (NH) winter season and a southern tilt in the NH summer, meaning the seasonality of the migration of the CE in the lower altitude is larger than that in the higher altitude. The meridional extent of the CE indicates a narrow transition zone where IHT happens throughout the year. We find that the meridional extent above South America is larger compared to other regions. The distribution of the land-sea contrast plays an important role in the meridional extent of IHT. We focus on the TWP region and further compared the tropical rain belt with the CE. There is a broad region of high precipitation occurring in the TWP region and it is difficult to determine the IHT by the rain belt. In the NH winter, the CE is not consistent with the tropical rain belt in the TWP but is confined to the southern branch of the peak of the rain belt. For the other seasons, both indicators of IHT agree.

## 1 Introduction

The Tropical West Pacific (TWP) is an area extending from the Maritime Continent (Ramage, 1968) to the International Date Line, with some of the world's highest sea surface temperatures. The TWP warm pool provides the environment in which

deep convective cloud systems develop (Fueglistaler et al., 2004). During Northern Hemispheric (NH) winter troposphere air ascends into the stratosphere via the Tropical Tropopause Layer (TTL) mainly in this region, the TWP is considered as the major transport pathway from the troposphere into the stratosphere during NH winter (Newell and Gould-Stewart, 1981; Fueglistaler et al., 2004; Krüger et al., 2008; Rex et al., 2014). Deep convection and large-scale ascent in this region enable boundary layer air heated by the warm ocean surface to ascend to the TTL, changing the composition of the TTL atmosphere

across the tropics. The air mass origin in the TWP region needs to be studied to properly describe the chemical species entering the stratosphere via the troposphere stratosphere transport pathway above the TWP.

    The Inter-Tropical Convergence Zone (ITCZ) is conventionally defined as a lower-tropospheric convergence region circling the globe where the tropical trade winds from the Northern and Southern hemispheres meet, typically lying between approximately 15°S and 15°N (Waliser and Gautier, 1993). In the TWP, the ITCZ migrates seasonally towards the hemisphere that

warms relative to the other (Schneider et al., 2014), indicating the migration of the circulation patterns in this region. Generally, it is characterized by fast vertical motion and heavy rainfall and essentially acts as a meteorological barrier to cross-equatorial flow. Previous studies recognize the ITCZ as the boundary to Inter-Hemispheric Transport (IHT) and/or interhemispheric mixing in the tropical region (Williams et al., 2002; Stehr et al., 2002). The location of the ITCZ affects weather conditions and air mass origin throughout the tropics. The time mean or the climatology of ITCZ locations can be characterized by zonal regions

of heavy rainfall in the tropics. The day-to-day features of the ITCZ, however, can be quite changeable over the landmasses and due to interactions with monsoon systems (Wang and Magnusdottir, 2006). The mechanisms controlling its position and rainfall intensity are not fully understood (Schneider et al., 2014).

    There are some disadvantages to using the ITCZ as an indicator of the equatorial circulation system. First, the ITCZ is difficult to define over the TWP region, because the Western Pacific Monsoon (WPM) (Smith et al., 2012) adds complexity to

the tropical rain belt. This broad region of the tropical rain belt makes it difficult to determine the location of the ITCZ. Second, the equatorial precipitation over land is not simply a response to the surface convergence but is also influenced by local factors such as convection caused by topography, proximity to the sea, and variation of the regional humidity. The assumption that the seasonal cycle of rainfall in equatorial Africa is controlled by the seasonal excursion of the ITCZ is therefore still challenged and discussed (Nicholson, 2009, 2018).

In the TWP region, Hamilton et al. (2008) first introduced the term Chemical Equator (CE) rather than the ITCZ to represent the main atmospheric boundary between the two meteorological hemispheres. They used CO as a tracer to investigate the CE and pointed out that this boundary between the two meteorological hemispheres does not coincide with the ITCZ or the monsoon trough. Methane ($CH_4$) and carbon dioxide ($CO_2$) have a relatively long lifetime and clear latitudinal gradient which has the potential as tracers to investigate IHT (Patra et al., 2011; Law et al., 2008; Lin and Rood, 1996). The atmospheric tracer

transport model intercomparison project (TransCom) investigated IHT by non-reactive tropospheric species such as $CO_2$, $CH_4$ and Sulfur hexafluoride ($SF_6$) and provided a comprehensive understanding of the differences in tracer distribution between the northern and southern hemispheres and studies IHT (Krol et al., 2018). $SF_6$ is a common tracer to constrain time scales of IHT (e.g., Geller et al., 1997; Waugh et al., 2013; Yang et al., 2019). It has a very long atmospheric lifetime (580 - 3200 yr) (Ravishankara et al., 1993; Morris et al., 1995; Ray et al., 2017), a large and constant growth rate during the last two decades

(Rigby et al., 2010; Hall et al., 2011) and anthropogenic sources primarily over the NH. However, tracers with significant north-south gradients (such as CO, $CH_4$ and $SF_6$) are locally diverse and can be affected by human activities. So, to eliminate the regional dependency on human activities and the chemical processes with other species, an artificial tracer without such features is needed to investigate IHT in the tropics.

This study aims to provide a tool to determine the boundary for air mass transport between the two meteorological hemi-
spheres in the tropics, focusing on the TWP region. Here, we present model simulations of a passive tracer to determine this boundary. Following Hamilton et al. (2008), we use the term CE to describe this boundary. This way, we avoid confusion with the tropical rain belt indicated by the conventional ITCZ definition. To assess regional differences only caused by air mass transport, we switched off the chemistry in the model to develop an atmospheric pattern only due to the transport by the analyzed wind fields. This way, we neglect chemical processes and regional dependency of the emissions occurring for real
species like CO and $SF_6$. Additionally, the model approach yields a three-dimensional pattern of air mass transport, which allows investigation of the vertical structure of interhemispheric mixing processes.

In Section 2, the model description with the simulation steps and setup are described. Section 2 also introduces the method to derive the location of the CE from the simulation results. In Sect. 3, the CE location results are shown in different seasons and regions. We compare the CE locations with the tropical rain belt and wind fields from reanalysis data and investigated
IHT in the tropics, especially in the TWP. We also compare the CE with the distribution of atmospheric compositions such as $CH_4$ and $SF_6$. We present the vertical and the meridional extent of the CE. In Section 4, we discuss the difference between the CE derived in this study and the ITCZ determined and presented in previous studies. The consistency between the CE and measurements of other trace gases such as ozone and $CH_4$ presented by previous studies are also discussed in Sect. 4. The discussion of the stability and uncertainty of this method is given in the Appendix.

## 80 2 Methods

### 2.1 GEOS-Chem Setup

We used the global 3-D chemical transport model GEOS-Chem in version 13.0.0 (Bey et al., 2001) driven by meteorology input from the Goddard Earth Observing System (GEOS) of the National Aeronautics and Space Administration (NASA) Global Modeling and Assimilation Office. The model was driven by the Modern-Era Retrospective analysis for Research
and Applications, Version 2 (MERRA-2) reanalysis meteorological fields produced by the Global Modeling and Assimilation Office (GMAO) at the Goddard Space Flight Center. The basic setup of our model simulation is summarized in Table. 1. We first used a coarse global simulation with a grid resolution of $2° \times 2.5°$ to determine the boundary conditions. Then, we performed nested simulations with the resolution of $0.5° \times 0.625°$ in the tropical zonal domain of 30°N to 30°S for the same period of the global simulation. The model runs used 72 vertical layers from the surface up to 10 hPa, and the output was saved
for every day. We switched off the chemistry in the model and emissions of compounds except for the passive trace, so only the advection is considered. The simulation results used in this study are therefore gridded $0.5° \times 0.625°$ horizontally in vertical 72 levels.

## 2.2 Determination of the Chemical Equator (CE)

### 2.2.1 Description of Different Experiments

A series of tracer experiments were made to investigate the CE. As shown in Table. 2, two base experiments, Experiment 1 (E1) and Experiment 2 (E2), are carried out to study the air mass transport from both hemispheres by releasing the tracer either in the northern or southern extra-tropics latitude bands (see Fig. 1). The simulation time of E1 and E2 is from 2014 to 2019 and we take the simulation in 2014 as a spin-up simulation. The tracer experiments follow the same format: the inert chemical tracers with infinite lifetime were released into the atmosphere with the constant flux of $1 \times 10^7$ kg/m$^2$/s from the start to the

end during each simulation. Similar to the actual vertical extent of the emission of the atmospheric component, the vertical extent of the emission of the passive tracer is from the surface to 1 km. The source domains of the passive tracer are marked by red and blue colors which means the passive tracer released from 30°N - 90°N and 30°S - 90°S, for different hemispheres, respectively.

     The uniform flux in the zonal range within the extra-tropics was chosen to eliminate the regional dependency on emissions.

After release, the tracer first accumulated in the according hemisphere of release and then travelled to the other hemisphere, resulting in a stable north-south gradient pattern as a result of air mass transport and atmospheric circulation. For the sake of simplicity and clarity, the methodology of determining the CE introduced after is described based on E1. In the base experiment E1, the tracer was released in the NH extra-tropics to determine the northern boundary of the CE, which is abbreviated as CE-NH. The setup of the E2 was the same as that of the E1, except that the passive tracer emission region was placed in the SH.

With the same method but applied to the simulation results from the E2, we can obtain the southern boundary of the CE called CE-SH.

     Fig. 2a shows a time series of the global distribution of the tracer averaged zonally in six latitude bands for the releasing of the tracer from 30°N - 90°N. After approximately one year of simulation, the linear growth rate is roughly equal in each latitude band. The meridional gradient of the passive tracer is similar to SF$_6$ shown in Fig. 2b, supporting the use of a passive

tracer for the study of IHT.

     Apart from these two base experiments, there are other experiments Experiment 3 (E3) - Experiment 5 (E5) designed to investigate the stability of the method and to ensure that it is robust in different model settings. These experiments and the results are described in more detail in the Appendix A, and the basic setups are summarized in Appendix A and A1. Releasing tracers on different altitude ranges does not affect the method. For example, the uniform release of the tracer between the

surface and 10 km only affects the spin-up time of each experimental case, not the distribution of the tracer on the ground. This method can also be used to determine the location of CE in other years of interest like E5 (simulating from 2010 to 2015), which differs from the simulation starting time of E1 - E4, indicating good repeatability of this method.

### 2.2.2  Decomposition Method

To distinguish air mass transport from either one or the other hemisphere, the decomposition method is applied to the time series of the tracer, thus deriving the trend and the seasonality. An additive model of the decomposition is used:

$$y_t = T_t + S_t + R_t, \tag{1}$$

where $y_t$ is the time series of the tracer, $T_t$ is the trend component, $S_t$ is the seasonal component, $R_t$ is the residual component or noise. The subscript $t$ denotes the time.

Fig. 3 shows the decomposition of the time series of the passive tracer released from 30°N - 90°N. Two grid boxes in the TWP are shown here as examples, one located at 6.0°S, 127.5°E and the other located at 6.0°N, 127.5°E. The trend component in each grid box is a linear increase. The seasonal component of the grid box in the NH (as shown in Fig. 3a) varied from positive values to negative values year-round, which shows the higher concentration from the higher latitude bands and lower concentration from the lower latitude bands. For the grid box in the SH (shown in Fig. 3b), the seasonal component is positive only when a high concentration of the passive tracer is transported from the NH to this grid box. Otherwise, in other periods, approximately from March to November each year, the air mass from the south has a concentration value of zero, so the seasonal component is also around zero value.

After decomposition, the trend of the tracer in each grid box can be given as $T_{t,i,j}$ , where the subscript $t, i, j$ refer to the time, the longitude and the latitude of the grid box, respectively. The trend in each grid box and each time step are spatially averaged in the domain of -180°S to 180°and 30°S to 30°N:

$$\overline{T_t} = \overline{T_{t,i,j}},\, i \,\text{from} -180° \,\text{to}\, 180°,\, j \,\text{from}\, 30°\,\text{S to}\, 30°\,\text{N}, \tag{2}$$

where $\overline{T_t}$ is the spatial average of the trend which is also the criterion of the CE at each time step $t$.
The location of the CE in each time step $t$ is given by $\overline{T_t}$, which is the spatial average at each time step of the trend, which indicates the tracer concentration on the CE:

$$CE_t = \left\{\text{where}: C_{i,j,t} = \overline{T_t}\right\}, \tag{3}$$

where $C_{i,j,t}$ is the tracer concentration in each grid box and each time step $t$ . For example, if the concentration of the tracer (released from 30°N - 90°N) in a grid box is higher than $\overline{T_t}$ , this grid box is located on the NH, and vice versa for the SH.

The CE-NH and CE-SH calculated by the decomposition method and the global distributions of the passive tracer averaged in January at each year of the simulation time from 2015 to 2019 are shown in the Fig. 4. The concentration of the passive tracer gradually increases after the releasing time in both experiment cases. This latitudinal gradient can be clearly seen in the distribution of the passive tracer and is well determined by the CE-NH and CE-SH. There is another common used method to determine the CE or ITCZ from the gradient of the tracer. The comparison of our decomposition method and the gradient method is given in Appendix B, and it shows a less robust results by the gradient compared to the decomposition method.

By comparing the results of the two base experiments E1 and E2, we obtain insights into IHT and answer an important question of whether the northern and southern boundaries of the CE coincide with each other when the passive tracer was

released in different latitude bands coming from two different hemispheres. The region between these two boundary lines is where interhemispheric mixing happens and is referred to as the CE.

## 3   Results

### 3.1   The Interhemispheric Transport Indicated by the Chemical Equator

Figure. 5 shows the daily locations of the CE-NH and the CE-SH by colored scatters. Here, we only present the CE in 2015
as an example. For other years (2016 - 2019) similar distributions of the CE are given in the Supplement. Both CE-NH and CE-SH reach the southernmost position at the end of NH winter and the northernmost position at around $25°N$ in the end of NH summer, but do not coincide with each other (see Fig. 5a and Fig. 5b). In general, the daily CE-SH is further south than the CE-NH. In February, the CE-SH reaches around the southernmost position at $20°S$ in East Africa, the Indian Ocean, the Central Pacific, and South America. In late August, except for Africa and the Atlantic, the CE-NH reaches its northernmost
position at about $20°N$.

     The seasonal average of the CE from 2015 to 2019 is shown in Fig. 6. Here, the CE clearly shows as a belt of meridional extent around the tropics between the northern and the southern boundary CE-NH and CE-SH. Since atmospheric transport is a continuous process, we don't expect a single boundary line separating the atmosphere and matter in the NH and SH. The boundary is a belt of longitudinal width in which air masses from the NH exchange and mix with those from the SH. As shown
in Fig. 6a, from December to February, the CE is located south of the equator. After that, the CE moves north from March to August (shown in Fig. 6b and Fig. 6c), crossing the equator to reach the geographical NH. In the NH spring and summer, the progressive domination north of the equator by the air mass originating from the SH is characterized by the movement of the CE. In the NH autumn (Fig. 6d) and winter season (Fig. 6a) the air flows from the NH gradually strengthen and the boundary moves southward, finally reaching its southernmost position in NH winter (Fig. 6a). This suggests that the CE lags behind the
ground position of the sun by about 3 months and coincides with the time lag of the ITCZ.

     To further study the migration of the atmospheric boundary between the two hemispheres and its correlation to atmospheric circulation, we compare the circulation patterns over different regions. Figure 8 shows the annual movement of the CE and zonally-averaged 10-m wind vectors in different regions defined as rectangular boxes within the tropical band between $30°S$ and $30°N$ (see Fig. 7): Central & Eastern Pacific (CEP): $180°$- $80°W$; South America (SA): $80°W$ - $40°W$; Atlantic (AT):
$40°W$ - $15°W$; Africa (AF): $15°W$ - $50°E$; Indian Ocean (IO): $50°E$ - $100°W$; Tropical Western Pacific (TWP): $100°E$ - $180°$. The division of the regions is adapted from the definition of tropical regions by Fueglistaler et al. (2004). In general, the wind convergence zone is consistent with the latitude of the CE and follows a similar pattern: it is located south of the equator in winter and north in summer. But there are regional differences as described in the following. In the Central & Eastern Pacific and the Atlantic Ocean, the wind convergence zone agrees with the CE as shown in Fig. 8 (CEP) and Fig. 8
(AT). Here, throughout the year, north-easterly and south-westerly winds meet near the equator from $0°$ to $10°N$, forming a clear convergence zone, while the CE lies in the confluence bands of the winds. The annual movement of the CE is relatively

small in the Atlantic and Eastern Pacific, between 5°S and 10°N, indicating weaker seasonal shifts of the tropical circulation in those regions.

The seasonal movement of the CE is larger over the land sectors, i.e. tropical South America and Africa, and wind convergence zone and CE do not coincide, as shown in Fig. 8 (AF) and Fig. 8 (SA). In Africa, the confluence zone of north-easterly and south-westerly winds lies north of the CE. This implies that air masses from the NH are transported further south than the location of the wind convergence zone suggests. For South America, we cannot see a significant wind field convergence zone in Fig. 8 (SA). The contrast between the north-easterly winds from the NH and the South-easterly winds from the SH is obvious, possibly due to the distribution of land and sea. In NH winter, from December to February, the CE-SH reaches its overall southernmost position at 15°S in South America.

The circulation system in the TWP and its interaction with the large-scale atmospheric circulation such as WPM and Hadley cell bring much complexity to the studies in this region. Over the TWP and the Indian Ocean, the annual movement of the CE is larger than over other ocean sectors, such as the Atlantic Ocean and Central & Eastern Pacific. From December to April, north-easterly winds deflect west after crossing the equator and converge with south-easterly winds from the SH between about 0° and 5°S. From May to November, this convergence zone moves northward, while the south-easterly winds turn west after crossing the equator and converge with north-easterly winds north of the equator.

### 3.2   The Chemical Equator and Atmospheric Composition

To better understand the implication of the CE position on air composition, satellite measurements of $CH_4$ and model simulation of $SF_6$ are presented together with the CE in Fig. 9. Column-averaged dry air mole fractions of $CH_4$ ($XCH_4$) radiance measurements in the shortwave infrared (SWIR) bands (2305 - 2385 nm) of the TROPOspheric Monitoring Instrument (TROPOMI) aboard the Sentinel-5 Precursor satellite mission (Veefkind et al., 2012). Here we use the latest release of the WFMD (Weighting Function Modified Differential Optical Absorption Spectroscopy) product (v1.8) (Schneising et al., 2023) and process it onto a $2° \times 2°$ grid. The details of the satellite data product are described in Appendix C. We used GEOS-Chem v13.0.0 to obtain the $SF_6$ distribution. The model setup of $SF_6$ is described in detail in Appendix D. The CE and the global distribution of $CH_4$ and $SF_6$ averaged for January and July 2019.

The CE and the north-south gradient of $XCH_4$ in the Indian Ocean and Eastern Pacific in January and Africa in July are well consistent with each other. This indicates the CE has good potential to illustrate the IHT inferred by observations. However, due to the lack of data coverage, it is relatively difficult to see the distribution of $XCH_4$ in the SH in July. The $XCH_4$ distribution is also affected by sources in the SH and chemical removal process. This means the $XCH_4$ is not monotonically rising like the inert artificial tracer used in our study and does not show a clear distinction between NH and SH. For $SF_6$, there are large emissions in South East Asia, which may be emitted into the CE area, and the SH emission in South America, which may impact the latitudinal gradient of the $SF_6$. Apart from those emissions regions, the $SF_6$ distribution mostly shares the features of an artificial tracer, i.e. the monotonic rise, and therefore has been used for similar studies(e.g., Geller et al., 1997; Waugh et al., 2013; Yang et al., 2019).

### 3.3 Vertical Structure of the Chemical Equator

An advantage of our method of determining hemispheric boundaries is that it allows analysis of the vertical structure of the IHT. We calculate the CE for each vertical level of the model output. As Fig. 10 reveals, the CE-NH and CE-SH show less meridional variation at lower levels than at higher ones. In Fig. 10 we only present the vertical structure of the CE in the TWP (100°E - 180°, 30°S - 30°N, see Fig. 7). For other regions specified in Fig. 7, the vertical results of the CE are shown in the Supplement.

In general, the vertical sections of the hemispheric boundary of the atmosphere differ with seasons. From January to March, the CE tends to tilt north. During these three months, air masses from the NH below 2 km move south of the geographic equator to about 10°S, while air masses above 2 km south of the geographic equator originate in the SH. In April and May, the oblique structure becomes less pronounced and the CE begins to be vertical to the ground. From June to October, the CE tilts south. It should be noted that the CE-NH slopes southward from the ground up, while the CE-SH from the ground to 2 km is relatively uniform. This indicates that the airmasses originating in the SH near the ground do not move further north in the summer, but stay near the equator, forming a broader meridional mixing region in the boundary layer. In November, the CE-NH is the most uniform with altitude throughout the year and the overall CE is a more narrow band compared to other months. In December, the CE shows a slight sloping trend to the north. These two months, November and December resemble April and May with the most narrow and vertically uniform CE above 2 km throughout the year. They mark the turning points of IHT above 2 km altitude between a more northern and more southern position. In the discussion of the IHT we only take the model level under 8 km into consideration. With increasing altitude, the boundary between the two hemispheres is less pronounced due to the fast horizontal mixing by high-speed winds in the upper troposphere.

### 3.4 The Chemical Equator Versus the Tropical Rain Belt in the TWP Region

The rain belt is usually regarded as an indicator of the equatorial convergence zone and thus the boundary for interhemispheric exchange. However, the rain belt in the TWP region is relatively complex due to the convergence of the commonly defined ITCZ and its annual movement. Here, we compare the rain rate with the CE in the TWP region: Figure 11 shows the zonal rain rate averaged from 2015 to 2019 and the results of the E1 and E2 simulation as a rate of occurrence, i.e. the number of days that the boundaries of the CE occur at each latitude as a percentage of the total appearance:

$$f_i = \frac{d_i}{\sum d_i}, \tag{4}$$

where $f_i$ is the rate, $i$ denotes the latitude, $d_i$ is the number of days that the CE-SH / CE-NH is located in the latitude $i$ . A higher rate of occurrence indicates a more frequent latitudinal position of the respective CE boundary.

In general, as shown in Fig. 11, the meridional range of the CE is more concentrated within a single month compared to the area encompassed by the rain bands. From May to October, in summer and autumn, the southern and northern peak of the rain belt coincide at the southern and northern boundaries of IHT, CE-SH and CE-NH, respectively, indicating that the location of the north-south rain belt during this time is related to actual air mass exchange between the two hemispheres.

Both CE-NH and CE-SH tend to be located at the southern peak of the rain band in the winter and early spring, from December to March. During these months the northern rain band is outside the range of the CE and therefore seems not associated with IHT. This suggests that the northern branch of the rain belt is related to the NH circulation system at around 5 - 10°N.

The seasonal cycle of both CE-NH and CE-SH is shown in Fig. 12 together with the rain rate in the TWP region. The meridional extent of the transition area between the NH and SH, i.e. the CE, varies with season. As already shown in the monthly results shown in Fig. 11, the CE is broader in NH summer than in other seasons. During NH winter (DJF, December–January) and spring (MAM, March-May), the CE is narrow and the northern part of the rain belt around 10°N is located north of the CE, while the southern part is included. This indicates, that the cause of the precipitation in the northern part of the TWP region is not the convergence of the equatorial flow from the NH and SH, but dependent on regional circulation within the NH. During NH summer (JJA, June-August) the meridional extent of the CE is the largest and includes the northern rain belt. During NH autumn (SON, September-November), the northern border, CE-NH, begins to retreat southward, and the CE again becomes more narrow again over the Maritime continent and coincides with the two rain belts in the NH and SH.

## 4 Discussion

The CE is in general not always in agreement with the pattern of the tropical rain belt as defined and analysed by previous studies (e.g., Adam et al., 2016; Schneider et al., 2014). The seasonal migration of the CE is more stable across the oceans than land, specifically in the East Pacific and the Atlantic. In these two regions, previous studies defined the convergence zone by either the tropical rain belt (Gu et al., 2005), low cloud-top temperature (Waliser and Gautier, 1993), or the magnitude of the horizontal gradient (Berry and Reeder, 2014).

The seasonal migration of the CE on the continents is tied to the higher complexity of the atmospheric circulation system compared to the ocean. The circulation is modulated by several regional features such as local atmospheric jets and waves, proximity to the oceans, terrain-induced convective systems, moisture recycling, and spatiotemporal variability of land cover and albedo, so the location of the tropical rain belt becomes diffuse and does not coincide with the atmospheric boundary of the hemispheres (e.g., Arraut et al., 2012; Dezfuli et al., 2017; Magee and Verdon-Kidd, 2018). As mentioned in Sect. 3.1, the meridional extent of the CE above the continents South America and tropical America is larger than the one near ocean sectors. Considering the land-sea distribution and the complexity of the circulation system over tropical continents, more studies are needed on the regional circulation in tropical continental regions.

Previous studies (Hamilton et al., 2008; Petersen et al., 2010; Zhou et al., 2018; Müller, 2020) based on trace gas observations by aircraft, ozone soundings and Fourier transform infrared (FTIR) spectrometers in the tropical regions were aimed to gain a better understanding of tropical dynamics. During NH winter, high concentrations of pollution tracers such as CO and Ozone from Southeastern Asia are transported towards the TWP by large-scale circulation, which is modulated by the migration of the ITCZ (Hamilton et al., 2008; Müller, 2020). This phenomenon was also captured by our method, which is shown in Fig. 12 (DJF). The region north of the CE-NH, which is at around 5°S, is considered as the meteorological NH. The FTIR

measurements at another tropical site, at Suriname, Paramaribo (5.8°N, 55.2°W), also suggest that the seasonal variation of $CH_4$ is highly related to IHT (Petersen et al., 2010). In Reunion Island (21°S, 55°E) a high spike of $CH_4$ coming from the NH was captured by the FTIR measurements in the local summer (December-February) (Zhou et al., 2018). As shown in Fig. 6, the CE is located around 20°S during this period, which is consistent with these observations. The consistent results of trace gas observations and our calculations of the CE for the tropical sites underline the potential of the CE as a good tool to determine

airmass origin and improve our understanding of tropical dynamics.

## 5 Conclusions

We introduced a new method to investigate interhemispheric air mass transport (IHT) in the tropical region by passive tracer simulations with GEOS-Chem. The so-called CE indicates the region where IHT occurs. Daily values of the CE show reasonable agreement with the pattern of the tropical rain belt. By comparing the CE with the wind field in different regions, we find

that the confluence of the equatorial flow is consistent with the CE where IHT occurs in the Central & Eastern Pacific and the Atlantic Ocean. In Africa, where the confluence zone is north of the CE, further investigations are needed. The vertical extent of the CE varies with the seasons. It slopes northward from the ground to higher altitudes in winter, is nearly perpendicular to the ground in the spring, and slopes southward in the summer. The tilt of the CE diminishes in the fall and returns to a pattern that is vertical to the ground. We focussed on the relationship between the CE and the tropical rain belt in the TWP region. The

north-south migration of the CE is not always consistent with the maximum rain rate during the year, especially in the TWP region.

Considering that air mass exchange is a continuous process, we performed simulations with a passive tracer release both in the NH and SH. Its extent varies with season and region. Two cases set in the two symmetry fluxes region in the NH and SH help to obtain a complete pattern of the IHT. This mixing process happens in a transition area, with a continuous gradient rather

than a single border separating the atmosphere in the NH and SH. From this transition area, we find that the northern part of the precipitation band in the TWP in winter is more likely caused by the regional circulation rather than the convergence of the equatorial flow from the NH and SH. By combining the CE determined from the two cases, we thus get further insights into the IHT in the TWP region.

The simulation results will be complemented by more observational data such as the ground-based observation network in

the future. Using the CE in combination with observations will allow a more detailed characterization of trace gas transport, sources and sinks in the TWP region. Since the TWP is an area of the active troposphere to stratosphere exchange, the seasonal and in particular vertical characteristics of the CE will be valuable for studies of troposphere to stratosphere exchange.

## Appendix A: Sensitivity study: Experiment 3 to Experiment 5

E3 to E5 are three supplement case studies with different emission regions and vertical layers compared to E1 and E2; the

setting of the simulations are shown in Table. A1. These cases are aimed to test the stability of our method to determine the

CE. It should be noted that the north-south gradient of this passive tracer is the initial condition of the method and the definition of the CE. So, the emissions of the tracer must exist continuously in one hemisphere to create and maintain this gradient, rather than an equilibrated atmosphere.

## Appendix B: CE by Gradient Method

The CE calculated from the latitudinal gradient of the passive tracer is shown in Fig. A3. In some regions, such as Easter Pacific and Atlantic Ocean, the gradient-based CE is consistent with the CE calculated by the trend as the method used in the main text of the study. But in general, the gradient-based CE is less stable than the trend-based CE in most areas, which shows a better potential to use the method based on trend to determine the CE than by the gradient. In some cases, e.g. Fig. A3b between -130 and -160 °E the gradient found by the steepest gradient does not make sense.

## Appendix C: CH$_4$ Products from TROPOMI


The Sentinel-5 Precursor satellite mission (Veefkind et al., 2012) was launched on 13 October 2017 carrying a single scientific instrument, TROPOMI, which is a nadir-viewing passive grating imaging spectrometer. The satellite is positioned in a near-polar, sun-synchronous orbit and has a swath width of 2600 km, which allows for daily coverage of the Earth. The retrieval is however dependent on sun-lit, cloud-free scenes which limits the daily coverage. The instrument consists of four spectrometers
measuring radiances in the ultraviolet, ultraviolet-visible, near-infrared, and short-wave infrared bands. XCH$_4$ used in this study is retrieved from TROPOMI measurements of sunlight reflected by Earth's surface and the atmosphere in the SWIR wavelengths (2300 - 2389 nm). The spatial resolution is $5.5 \times 7$ km$^2$ at nadir. The Weighting Function Modified Differential Optical Absorption Spectroscopy (WFMD) TROPOMI data product (Schneising et al., 2019) provides column-averaged dry air mole fractions of both CH$_4$ and CO. Here we use the latest release of the WFMD product (v1.8) (Schneising et al., 2023)
and process it onto a $2° \times 2°$ grid. For this, each measurement is assigned to a single grid cell and the weighted average of all measurements per cell is calculated. The measurements are weighted using the inverse standard deviation to disadvantage measurements with high uncertainty. Additionally, only measurements with a quality flag (qf) qf=0 (good) are included. Data coverage is therefore limited over regions with many clouds (e.g. tropics) or challenging measurement conditions.

## Appendix D: Model Set-up of the SF$_6$ Simulation

The meteorological fields used in the model are from MERRA-2 reanalysis as described in Sect. 2.1. We performed the simulation of SF$_6$ from 2014 to 2019 in the horizontal grid resolution of $2° \times 2.5°$ and vertical grid resolution of 72 levels. The emission database of SF$_6$ is annually spatially- girded and taken from the Emission Database for Global Atmospheric Research (EDGAR version 4.2) inventory (Muntean et al., 2018), available at $0.1° \times 0.1°$ global resolution for 1970 - 2008.

*Code availability.* The GEOS-Chem model code used in this analysis is downloaded from the GEOS-Chem "Science" Codebase repository: https://github.com/geoschem/geos-chem. The code for calculating the CE is available at https://github.com/XiaoyuSun-n/Chemical-Equator.

*Data availability.* The Meteorology input data used in this analysis are available at http://geoschemdata.wustl.edu/. The model output, the results of the CE, are available at: https://doi.org/10.5281/zenodo.7018391 (Sun et al., 2022).

*Author contributions.* XS and MP designed the study. XS conducted the model simulation and made the analysis. JH provided the TROPOMI $CH_4$ data and guidance. XS wrote the manuscript, with contributions from all coauthors. All authors discussed the results and commented on the manuscript.

*Competing interests.* The co-author Mathias Palm is a co-editor of ACP. Besides that, the contact author has declared that they don't have any other competing interests.

*Acknowledgements.* The authours acknowledge the GEOS-Chem Support Team at Harvard University and Dalhousie University for their effort. We also thank the support team in geos-chem github (issues). They gave us precious answers and resolved our doubts about GEOS-Chem models. This work has been supported by the BMBF (German Ministry of Research and Education) in the project ROMIC-II subproject TroStra (01LG1904A). The TROPOMI/WFMD version 1.8 data were obtained from the Institute of Environmental Physics, University of Bremen at https://www.iup.uni-bremen.de/carbon_ghg/products/tropomi_wfmd/ and have been generated using funding from ESA (GHG-CCI and MethaneCAMP projects, contract nos. 4000126450/19/I-NB and 4000137895/22/I-AG).

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

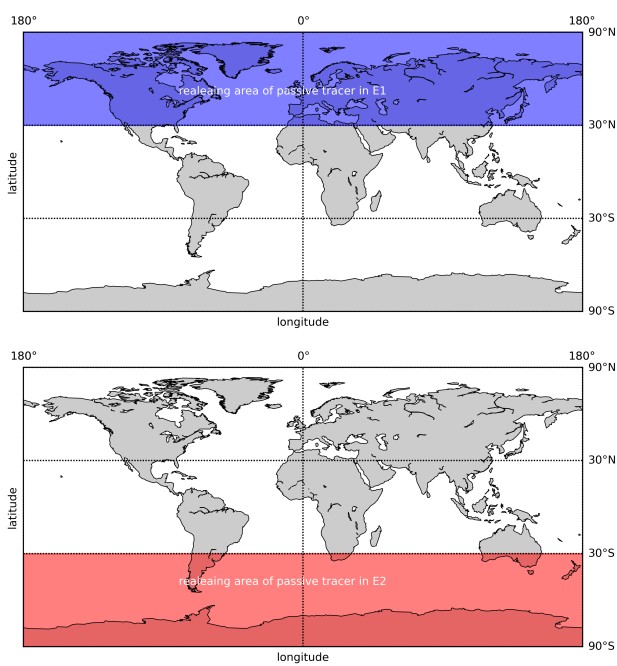

**Figure 1.** The releasing area of passive tracer E1 (shown by shaded blue region in the upper plot) and E2 (shown by shaded red region in the lower plot).

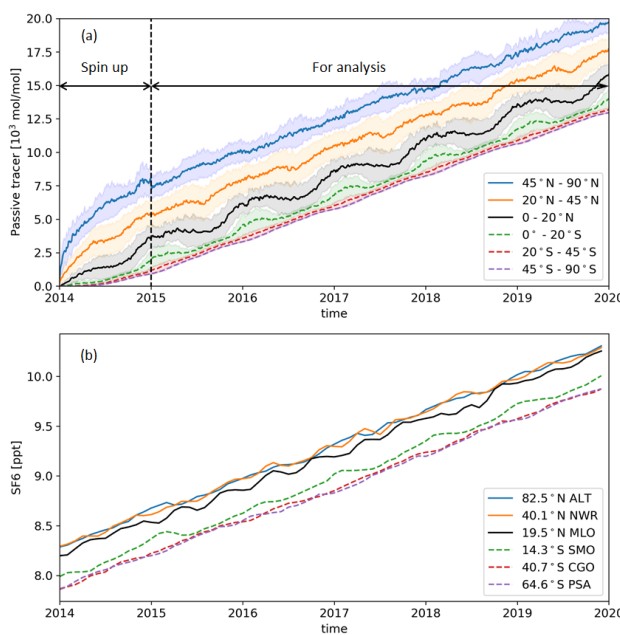

**Figure 2.** Comparison of the passive tracer and $SF_6$. (a) Zonally averaged amount of the passive tracer from GEOS-Chem simulations from 2014 - 2019 as a function of time for three northern (solid lines) and three southern (dashed lines) latitude ranges ($0°$ - $20°$, $20°$ - $45°$, $45°$ - $90°$). The value of the concentration of the passive tracer is not meaningful to the studies since we only take into account the relative higher or lower amount of the tracer. 1-$\sigma$ of the passive tracer of each latitude band is shown in shaded color. (b) $SF_6$ monthly means from Combined $SF_6$ data from the NOAA/ESRL Global Monitoring Division at six stations corresponding to the latitude bands in Fig.2a (ALT: Alert ($82.5°$N, $62.3°$W), NWR: Niwot Ridge ($40.1°$N, $105.6°$W), MLO: Mauna Loa ($19.5°$N, $155.6°$W), SMO: Cape Matatula ($14.3°$S, $170.6°$W), CGO: Cape Grim ($40.7°$S, $144.8°$E), PSA: Palmer Station ($64.6°$S, $64.0°$W)).

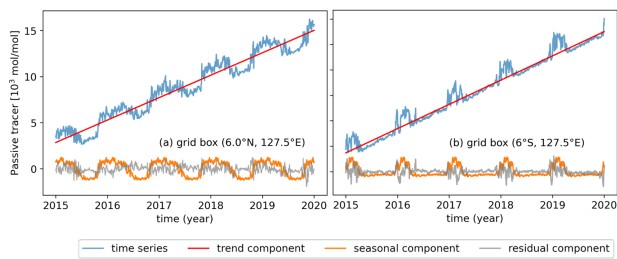

**Figure 3.** Time series of the passive tracer (blue line), trend component (red line), seasonal component (orange line), and residual component (grey line) of the passive tracer as a function of time (2015 - 2019) in two example grid boxes (a) [6.0°N, 127.5°E] and (b) [6.0°S, 127.5°E].

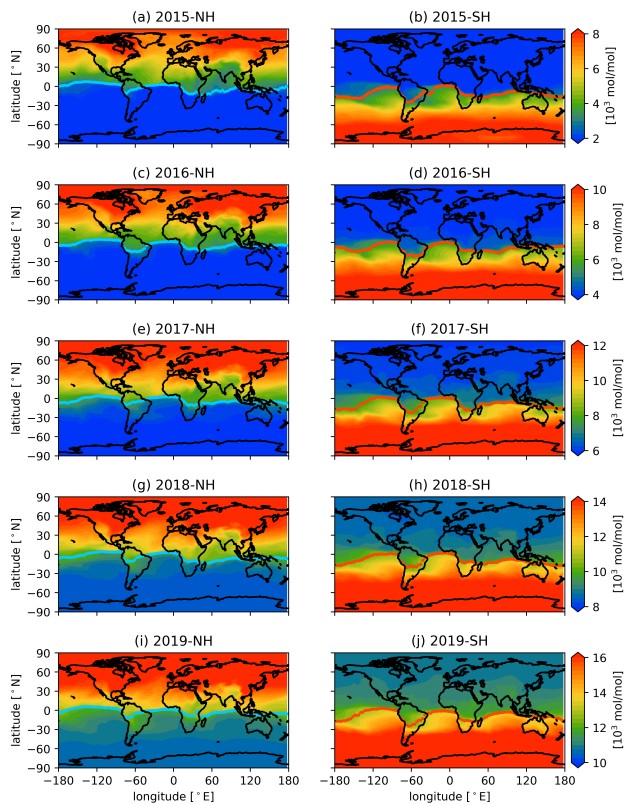

**Figure 4.** The surface concentration (mol/mol) of the passive tracer averaged in January at each year of the simulation from 2015 to 2019. The subplots in the left column (a), (c), (e), (g), (i) show the passive tracer released from the NH in Experiment 1 and subplots in the right column (b), (d), (f), (h), (j) show the passive tracer released from the SH in Experiment 2. The blue lines and the red lines show the CE-NH and CE-SH respectively.

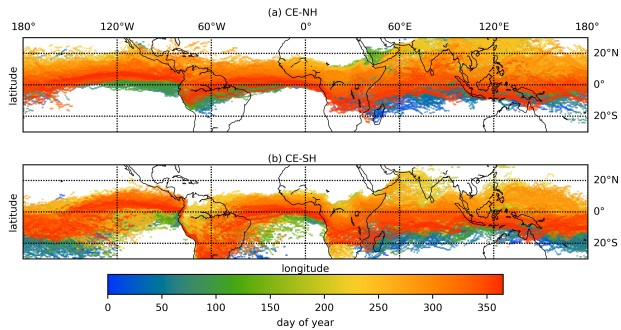

**Figure 5.** Daily CE-NH and CE-SH calculated from model simulations of (a) E1 (tracer released in NH) and (b) E2 (tracer released in SH) in 2015; the color shows the day of the year.

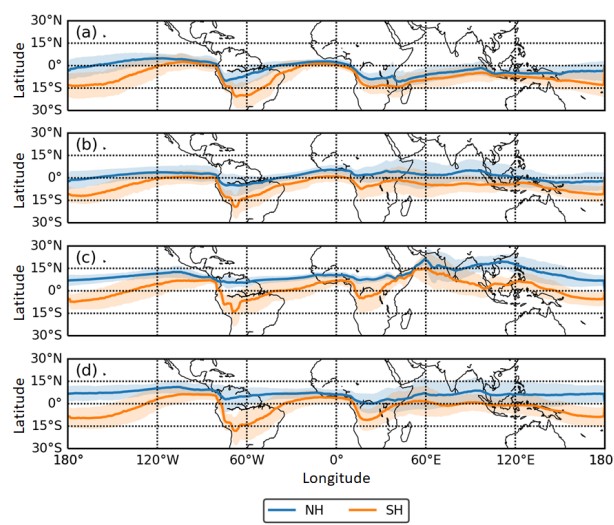

**Figure 6.** 5-year (2015 - 2019) averaged seasonal location of CE. (a) December, January, and February. (b) March, April, and May. (c) June, July, and August. (d) September, October, and November. 1-$\sigma$ of the CE-NH and CE-SH of each season is shown in shaded color.

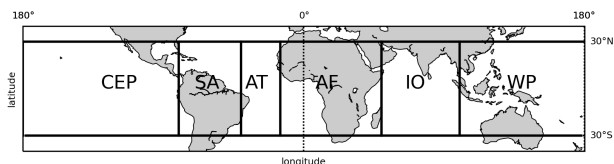

**Figure 7.** Definition of geographic regions in this study. Central & East Pacific (CEP): (180°, 80°W); South America (SA): (80°W, 40°W); Atlantic (AT): (40°W, 15°W); Africa (AF): (15°W, 50°E); Indian Ocean (IO): (50°E, 100°E); Tropical West Pacific (TWP): (100°E, 180°); all these regions are with the same latitude range: 30°S - 30°N.

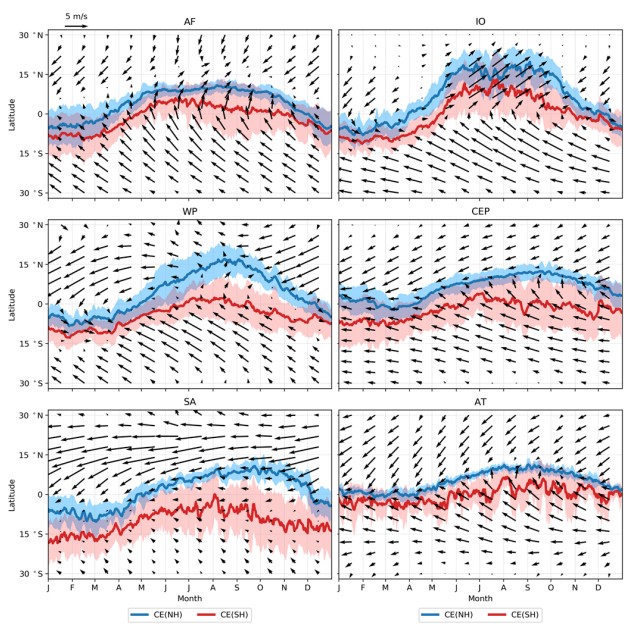

**Figure 8.** Monthly averaged wind vectors (black arrows) and annual movement of the daily CE. The blue lines show the NH boundary and the red lines show the SH boundary. The wind data are the 10-m winds from the ERA5 reanalysis data (Hersbach et al., 2020) averaged from 2015 to 2019. Both the CE and the wind field are space averaged zonally in eight different regions such as Africa (AF) and IO (Indian Ocean). The abbreviations and definition of the region on the top of each subplot are according to Fig. 7. 1-$\sigma$ of the CE-NH and CE-SH is shown in shaded color.

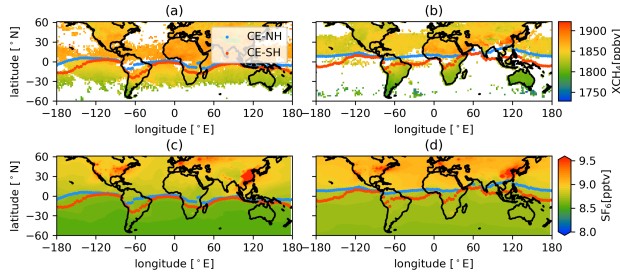

**Figure 9.** CE with Sentinel-5 Precursor satellite $XCH_4$ vertical columns (ppbv) averaged for (a) January 2019 and (b) July 2019. CE with $SF_6$ surface concentration (ppbv) simulated by GEOS-Chem averaged for (c) January 2019 and (d) July 2019. The blue dots show the NH boundary and the red dots show the SH boundary.

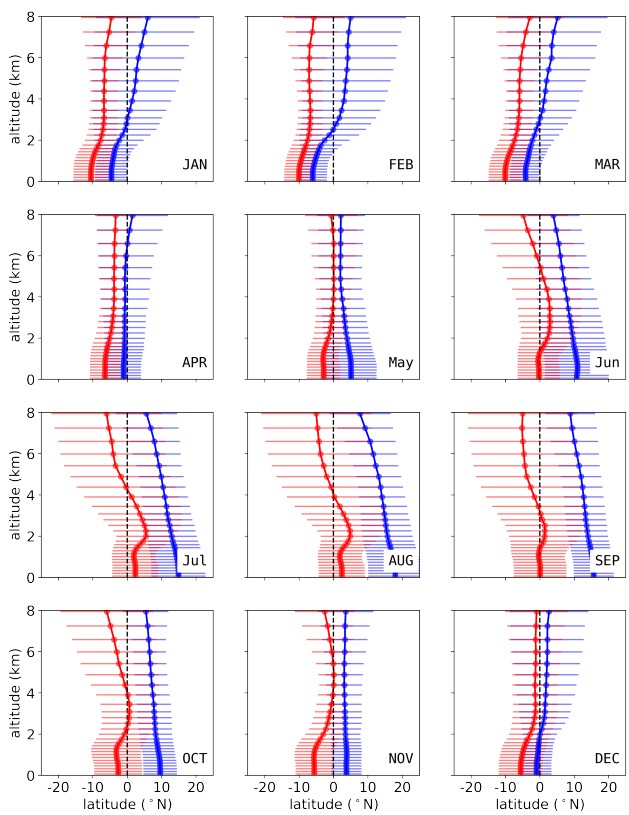

**Figure 10.** Monthly averaged (2015 - 2019) CE at different model levels from surface to 8 km. The CE-SH / CE-NH are zonally (100°E-180°) averaged over the TWP region see Fig. 7. The blue lines show the CE-NH and the red lines show the CE-SH. The dashed black line shows the latitude =0. 1-$\sigma$ of the CE-NH and CE-SH are given as thin horizontal lines in respective colours.

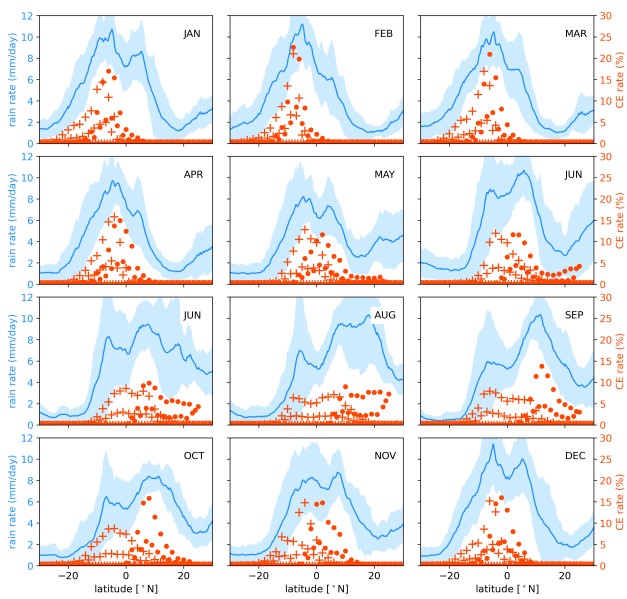

**Figure 11.** 5-year averaged (2015 - 2019) monthly rate of the CE-SH and CE-NH (red) with the rain rate (blue) from TRMM (Tropical Rainfall Measuring Mission) products 3B43 (monthly) (Huffman et al., 2007) as a function of latitude averaged over the West Pacific region (same definition as Fig. 7). The CE-SH is marked by '+' and the CE-NH is marked by dots.

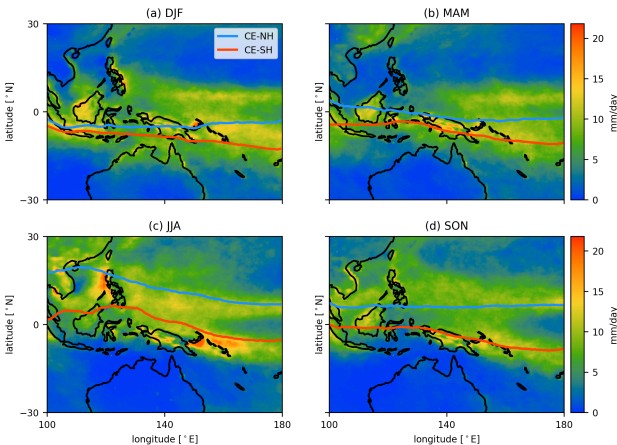

**Figure 12.** Seasonal rain rate (color scale) from TRMM (same dataset as Fig. 11) in the TWP region with the blue line showing CE-NH and the red line showing CE-SH. NH winter: December-February, DJF, NH spring: March-April, MMA, NH summer: June-August, JJA, NH autumn: September-November, SON.

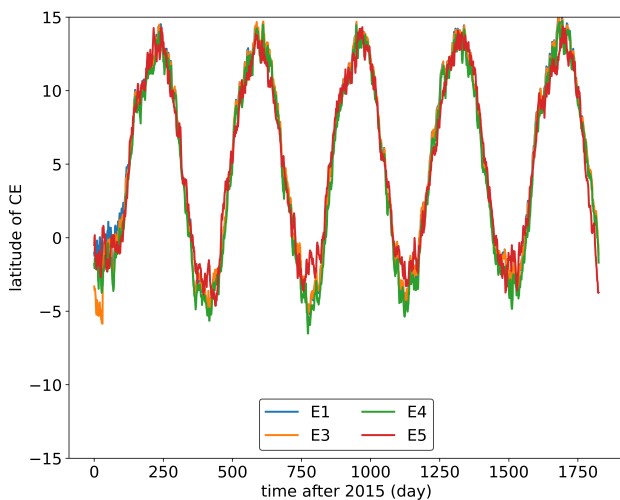

**Figure A1.** Comparison of the CE-NH in the basic experiment E1 with experiments E3 to E5. 5-year (2015 - 2019) zonally averaged daily latitude of all CE-NH.

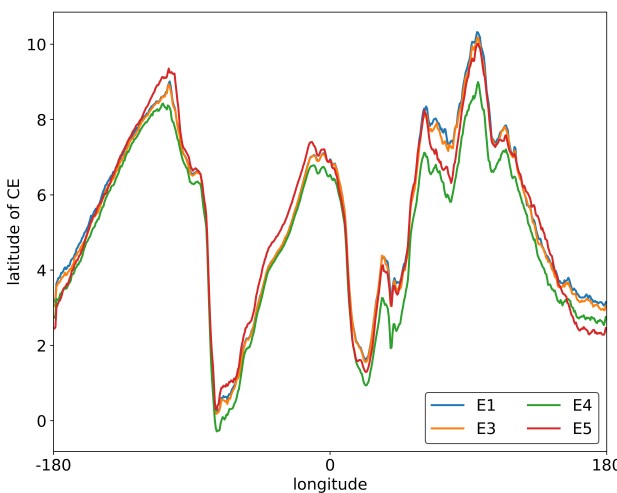

**Figure A2.** Comparison of the CE-NH in the basic experiment E1 with experiments E3 to E5. 5-year (2015 - 2019 or 2011 - 2015) daily latitude of all CE relative to longitude.

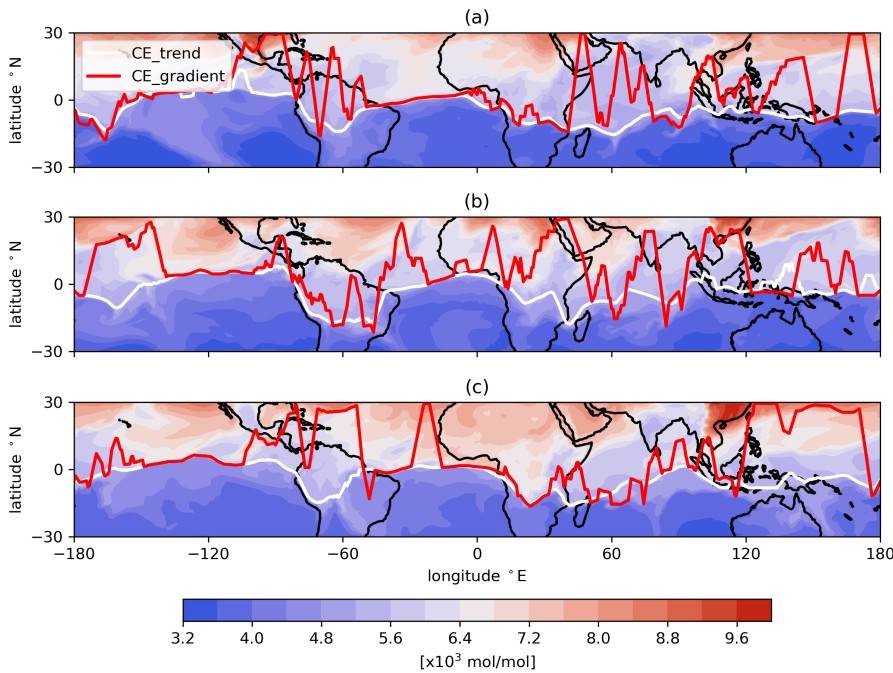

**Figure A3.** The CE which is calculated by the trend (CE_{trend}, solid black line) compares to the CE which is calculated by the latitudinal gradient of the passive tracer (CE_{gradient}, dashed dot line). The upper, middle, and lower plots are CE with the distribution of the passive tracer on 1, 15, and 31 January 2016.

**Table 1.** The settings of GEOS-Chem v13.0.0 used in this study

| | |
|---|---|
| Resolution | $2° \times 2.5°$ (Global simulation), $0.5° \times 0.625°$ (Nested simulation), 72 levels |
| Simulated species | Passive tracer |
| Global inventory | none |
| Meteorology field | MERRA-2 |
| Tracer lifetime | Infinite |
| Tracer emission | Constantly released after starting the simulation |
| Tracer chemical process | none |
| Transport / Convection timestep | 600 s |

**Table 2.** The settings of two base experiments

| Experiment | Release area | Release layer | Simulated time |
|---|---|---|---|
| E1 (NH) | 30°N - 90°N, zonally | Surface - 1 km | years (2014 - 2019) |
| E2 (SH) | 30°S - 90°S, zonally | | |

**Table A1.** The settings for experiment E3 to experiment E5

| Experiment | Release area | Release layer | Simulated time |
|---|---|---|---|
| E3 | 30°N - 70°N, -180°- 180° | Surface - 1 km | 5 years (2014 - 2019) |
| E4 | 30°N - 90°N, -180°- 180° | Surface - 10 km | 5 years (2014 - 2019) |
| E5 | 30°N - 90°N, -180°- 180° | Surface - 1 km | 5 years (2010 - 2015) |