# Peer review of "Determination of the chemical equator from GEOS-Chem model simulation: a focus on the Tropical Western Pacific region"

_Atmospheric Chemistry and Physics, 2022_

## Referee Comment (RC2)

Review of the paper:

"Determination of the chemical equator from GEOS-Chem model..."

written by Xiaoyu Sun et al.,

**General Comments:**
The paper presents a method to determine the hemispheric boundary of air mass transport between the two hemispheres called the chemical equator, which has not been topic of many studies so far. Particularly, it focuses on the Tropical West Pacific region. The location of chemical equator is calculated from the artificial tracers simulated by the GEOS-Chem model. The authors investigate the vertical structure of the chemical equator and compare the chemical equator to the tropical rain belt and the convergence of the wind fields. The topic of this paper falls into the scope of ACP. However, major revisions are required to the manuscript before it is suitable for publication. As described below, there needs to be (1) improved justification of the method to determine the chemical equator, (2) clarification of which vertical level are the chemical equator results based on, (3) application of the chemical equator defined here in understanding the inter-hemispheric transport of air, and (4) more proof to support the interpretation of some results.

**Major comments:**

- Previous study (Hamilton et al., 2008) used a sharp gradient in the chemical background to determine the chemical equator. This study defines the chemical equator as the location where the tracer concentration is the same to the average of the trend component over the tropics (30° S-30° N). Did the authors compare the chemical equator determined here to that calculated from the gradient of tracers?

  Hamilton, J. F., et al. (2008), Observations of an atmospheric chemical equator and its implications for the tropical warm pool region, J. Geophys. Res., 113, D20313, doi:10.1029/2008JD009940.

- The chemical equator can be obtained from the surface to the tropopause. It is unclear which vertical level are the results in the paper based on. Please check the whole manuscript from the method to the results and clarify.

- Since it is the chemical equator, it is good to know the connection between the location of the chemical equator and the distribution of atmospheric compositions (e.g. CO, $SF_6$) using satellite observations or model output. It will help to understand the inter-hemispheric transport of pollutants considering the difference of the tracer distribution in northern and southern hemisphere.

- The Western Pacific Monsoon and other circulations make the Tropical Western Pacific region complex. Except the large seasonal migration, what is the unique feature of the chemical equator over the Tropical Western Pacific comparing to that over the other regions. There needs be more discussions of chemical equator over the Tropical Western Pacific versus that over the other regions.

- There are a lot of interpretations regarding the air mass transport without relevant plots. It might be easier to understand the relative contributions from the source domains and air mass inter-hemispheric transport if the authors include the distribution of the artificial tracers from 30° N - 90° N and 30° S - 90° S.

**Minor and technical comments:**

- P1 L12
  "......meaning the speed of the migration of the CE decreases with the altitude", do you mean the seasonality of the migration of the CE in the lower altitude is larger than that in the higher altitude?

- P2 L47
  "......discussed. (Nicholson, 2009, 2018)." → "......discussed (Nicholson, 2009, 2018)."

- P3 L88
  "......zonal range of 30° N - 90° N..." → "......zonal range of 30° N - 90° N and 30° S - 90° S..."

- P4 L92
  "The time series of the tracer averaged zonally..." → "The time series of the tracer released from 30° N - 90° N averaged zonally..."

  Figure presents two source domains.Please specify the source domain to avoid the confusion.

- P4 L98-L99
  "Since the variation around the trend does not vary with the level of the time series", what do you mean here? What is "the level of the time series"? Do you mean different time period or different vertical level? If it is different time period or altitude, I don't understand why the variation around the trend does not change.

- P4 L103
  "Figure 3 shows the decomposition of the time series of the passive tracer." → "Figure 3 shows the decomposition of the time series of the passive tracer released from 30° N - 90° N."

- P4 L112
  "he longitude and attitude..." → "he longitude and latitude"

- P4 L112
  "The trend in each grid box and each time step is meridional averaged in the longitude of range of -180° to 180° and zonally averaged in the latitude range of 30° S to 30° N". It seems like this sentence might be "The trend in each time step is zonally averaged in the longitude of range of -180° to 180° and meridionally averaged in the latitude range of 30° S to 30° N"

- P4 L120-121
  "For example, if the tracer concentration in a grid box is higher than $\overline{T_t}$ , this grid box is located on the NH, and vice versa for the SH." The authors should mention this is for the tracers originated from 30° N - 90° N.

- P5 L149
  ”Figure 4 shows the daily locations of the CE-SH and CE-SH...” → ”Figure 4 shows the daily locations of the CE-NH and CE-SH...”

- P6 L156-L157
  ”The north of the boundary at 20° S is dominated by the airmass transport from NH”. I'm not sure about this interpretation, especially for the region from 20° S to equator. Do you have the results about airmass distribution?

- P6 L157
  ”...except Africa...” → ”...except Atlantic and Africa...”

- P6 L174-L177
  It is better to introduce Figure 6 before Figure 7.

- P6 L178
  ”...in the literature of (Fueglistaler et al., 2004)” → ”...in the literature of Fueglistaler et al. (2004)”

- P6 L186-L87
  ”In the NH winter, ... in boreal winter from December to February.” repetitive winter in one sentence.

- P7 L203
  ”the CE-SH and CE-SH” → ”the CE-NH and CE-SH”.

- P7 L209-L210
  ”less than 2 km” → ”below 2 km”.
  ”while the area above...”, it is better to specify the latitude range.
  ”From April” → ”In April and May”

- P7 L212
  ”while CE-SH slopes from the ground to 2 km and is relatively vertical to the ground”, it might be better to change as ”while CE-SH from the ground to 2 km is relatively vertical to the ground”

- P7 L217-L218
  Like the question I addressed in the abstract, how to understand the movement speed of CE here? Is it monthly movement speed?

- P8 L228
  ”the CE-SH / CE-SH...” → ”the CE-SH / CE-NH”.

- P10 L289-L290
  ”The north-south migration of the CE is consistent with the maximum rain rate during a year.” It is not always consistent, especially in JJA (Figure 10).

- P17
  Figure 3(b): legend (6° S, 127.5° E)
  Caption: (a) [6.0° N, 127.5° E] and (b) [6.0° S, 127.5° E].

- P22
  It is better to include a vertical dashed line along latitude = 0 in Figure 8.

- P23
  Caption: This sentence "The data in this plot are also zonally averaged in the TWP region specified in Fig. 5." is not necessary since "averaged over the West Pacific region" was mentioned before in the caption.

- P24
  Please include coordinate in Figure 10.

**Recommendation**

Read the manuscript thoroughly. Further improvements to the text clarity is necessary.

---

## Author Comment (AC1)

Response to Comments of Reviewer 1

*The authors thank all reviewers for their constructive comments and suggestions, which have helped us to improve the quality of this paper both in sciences and writing. All comments are carefully considered and responded to. The response in blue italic letters follows each comment in black.*

General Comments:

The paper presents a method to determine the hemispheric boundary of air mass transport between the two hemispheres called the chemical equator, which has not been topic of many studies so far. Particularly, it focuses on the Tropical West Pacific region. The location of chemical equator is calculated from the artificial tracers simulated by the GEOS-Chem model. The authors investigate the vertical structure of the chemical equator and compare the chemical equator to the tropical rain belt and the convergence of the wind fields. The topic of this paper falls into the scope of ACP. However, major revisions are required to the manuscript before it is suitable for publication. As described below, there needs to be (1) improved justification of the method to determine the chemical equator, (2) clarification of which vertical level are the chemical equator results based on, (3) application of the chemical equator defined here in understanding the inter-hemispheric transport of air, and (4) more proof to support the interpretation of some results.

Major comments:

• Previous study (Hamilton et al., 2008) used a sharp gradient in the chemical background to determine the chemical equator. This study defines the chemical equator as the location where the tracer concentration is the same to the average of the trend component over the tropics (30° S-30° N). Did the authors compare the chemical equator determined here to that calculated from the gradient of tracers?

Hamilton, J. F., et al. (2008), Observations of an atmospheric chemical equator and its implications for the tropical warm pool region, J. Geophys. Res., 113, D20313, doi:10.1029/2008JD009940.

*Response: We added an Appendix with the following Fig.1 to show the CE-NH determined by the latitudinal gradient of the passive tracers. CE-NH calculated from the latitudinal gradient of the passive tracer released from the NH (30 °N-30 °S) is shown in Fig.1. In some regions, such as the Eastern Pacific and the Atlantic Ocean, the gradient-based CE-NH is consistent with the CE-NH calculated by the trend as calculated by our method. But in general, the gradient-based CE-NH is less stable than the trend-based CE-NH in most areas, which shows a better potential to use the method based on trend to determine the CE-NH than by the gradient. In some cases, e.g. 1b between -130° and -160° E the gradient found by the steepest gradient does not make sense.*

[Figure]

*Figure 1. The CE-NH which is calculated by the trend (CE_trend, white line) compares to the CE-NH which is calculated by the latitudinal gradient of the passive tracer (CE_ gradient, red line). The upper, middle, and lower plots are CE-NH with the surface concentration of the passive tracer (mol/mol) on (a) 1, (b) 15, and (c) 31 January 2016.*

• The chemical equator can be obtained from the surface to the tropopause. It is unclear which vertical level are the results in the paper based on. Please check the whole manuscript from the method to the results and clarify.

*Response: The results in the paper are based on the vertical levels of GEOS-Chem which are 72 layers from the surface up to 10 hPa / 80 km. We checked the whole manuscript and made the following changes:*

- *We added the sentence:*

*"The simulation results used in this study are based on the vertical and horizontal grids which are 72 levels and 0.5°x0.625°, respectively."*

- *And we corrected the sentence:*

*"We calculate CE at each vertical grid of the model output of the passive tracer. With increasing altitude, it becomes hard to find an actual boundary between the two hemispheres due to the fast horizontal mixing by high-speed winds in the upper troposphere and lower stratosphere. So, we only take the model level under 8 km into consideration."*

- *And we corrected the caption of figure 9 (in the original manuscript) like this:*

*"Figure 9. Monthly averaged (2015-2019) CE at different model levels from surface to 8 km. The CE-NH / CE-SH are zonally (100° E-180°) averaged over the TWP region see Fig. 6. The blue lines show the CE-NH and the red lines show the CE-SH. 1-σ of the CE-NH and CE-SH are given in the plots.* "

• Since it is the chemical equator, it is good to know the connection between the location of the chemical equator and the distribution of atmospheric compositions (e.g. CO, $SF_6$) using satellite observations or model output. It will help to understand the inter-hemispheric transport of pollutants considering the difference of the tracer distribution in northern and southern hemisphere.

*Response: Thanks for the suggestion. We added the comparison of the CE with the global distribution of vertical columns of $CH_4$ from TROPOMI and the surface concentration of $SF_6$ from the GEOS-Chem simulation.*

*We added Sect. 3.2 to describe the connection between the CE and the distribution of $CH_4$ and $SF_6$:*

*"Section 3.2 The Chemical Equator and the distribution of atmospheric compositions*

*To better understand the implication of the CE position, satellite measurements of $CH_4$ and model simulation of SF6 are presented together with the CE in Fig. 2 . $CH_4$ used in this study is retrieved from TROPOMI measurements aboard in Sentinel-5 Precursor satellite mission (Veefkind et al., 2012) in the SWIR wavelengths (2300-2389 nm). Here we use the latest release of the WFMD (Weighting Function Modified Differential Optical Absorption Spectroscopy) product (v1.8) (Schneising et al., 2023) and process it onto a 5° x 5° grid. The details of the satellite data product is described in the Appendix C. We used GEOS-Chem v13.0.0 to obtain the simulation of $SF_6$. The model set-up of $SF_6$ is described in details in Appendix D. The CE and the global distribution of $CH_4$ and $SF_6$ averaged for January and July 2019 are shown in Fig. 2. The CE and the north-south gradient of $CH_4$ in the Indian Ocean in January 2019 are well consistent with each other. This indicates the CE has good potential to illustrate the IHT inferred by the satellite measurements of $CH_4$. However, due to the lack of data coverage, it is relatively difficult to see the distribution of $CH_4$ in SH from the satellite measurement in July. The $CH_4$ distribution is also affected by 1) sources in the SH and b) removal due to OH. This means the $CH_4$ concentration is not monotonically rising like the inert artificial tracer used in our study and does not show a clear distinction between NH and SH. $SF_6$ has this property and has been used for similar purposes (e.g., Geller et al., 1997; Waugh et al., 2013; Yang et al., 2019), but there are large emissions in South East Asia, which may be emitted into the CE area."*

*And we added Appendix to describe the satellite products of $CH_4$ we use and the GEOS-Chem model set-up of the $SF_6$.*

*"Appendix C*

*The Sentinel-5 Precursor satellite mission (Veefkind et al., 2012) was launched on 13 October 2017 carrying a single scientific instrument, TROPOMI, which is a nadir viewing passive grating imaging spectrometer. The satellite is positioned in a near-polar, sun-synchronous orbit and has a swath width of 2600 km, which allows for daily coverage of the Earth. The retrieval is however dependent on sun-lit, cloud-free scenes which limits the daily coverage. The instrument consists of four spectrometers measuring radiances in the ultraviolet, ultraviolet-visible, near-infrared, and short-wave infrared bands. $CH_4$ used in this study is retrieved from TROPOMI measurements of sunlight reflected by Earth's surface and the atmosphere in the SWIR wavelengths (2300-2389 nm). The spatial resolution is 5.5x7 $km^2$. The Weighting Function Modified Differential Optical Absorption Spectroscopy (WFMD) TROPOMI data product (Schneising et al., 2019) provides vertical columns of both methane $CH_4$ and carbon monoxide. Here we use the latest release of the WFMD product (v1.8) (Schneising et al., 2023) and process it onto*

*a 5° x 5° grid. For this, each measurement is assigned to a single grid cell and the weighted average of all measurements per cell is calculated. The measurements are weighted using the inverse standard deviation to disadvantage measurements with high uncertainty. Additionally, only measurements with a quality flag (qf) qf=0 (good) are included. Data coverage is therefore limited over regions with many clouds (e.g. tropics) or challenging measurement conditions.*

*Appendix D*

*The meteorological fields used in the model are from MERRA-2 reanalysis as described in Sec. 2.1. We performed the simulation of SF$_6$ from 2014 to 2019 in the horizontal grid resolution of 2° x 2.5° and vertical grid resolution of 72 levels. The emission database of SF$_6$ is annually spatially- girded and taken from the Emission Database for Global Atmospheric Research (EDGAR version 4.2) inventory (Muntean et al., 2018), available at 0.1° x 0.1° global resolution for 1970-2008.*

*"*

*And we added the following Figure 2 in the future revised manuscript:*

[Figure]

*Figure 2. CE with Sentinel-5 Precursor satellite CH$_4$ vertical columns (ppbv) averaged for (a) January 2019 and (b) July 2019. CE with SF$_6$ surface concentration (ppbv) simulated by GEOS-Chem averaged for (c) January 2019 and (d) July 2019. The blue dots show the NH boundary (CE-NH) and the red dots show the SH boundary (CE-SH).*

• The Western Pacific Monsoon and other circulations make the Tropical Western Pacific region complex. Except the large seasonal migration, what is the unique feature of the chemical equator over the Tropical Western Pacific comparing to that over the other regions. There needs be more discussions of chemical equator over the Tropical Western Pacific versus that over the other regions.

*Response: The CE is a tool to determine the boundary for air mass transport on a global scale. So the boundary determined by the CE basically has no unique feature in the TWP region compared to other regions over the tropics, despite the complicated circulation pattern. This is the reason why the use of the ITCZ fails to clearly separate the hemispheres in the TWP. But our study and others using models (Hamilton et al., 2008) show that such a separation exists.*

*However, the reason we choose to focus on the TWP region is that this region is considered as the major transport pathway from the troposphere into the stratosphere during the NH winter. So the air mass transport and origins in this region are valuable to be studied in detail, and therefore the main application of the CE in this study is to investigate the air mass transport in the TWP. But the method*

*of the CE can also be used for similar studies in other parts of the world, i.e. Africa or South America, which also show a complicated circulation due to the orography.*

• There are a lot of interpretations regarding the air mass transport without relevant plots. It might be easier to understand the relative contributions from the source domains and air mass inter-hemispheric transport if the authors include the distribution of the artificial tracers from 30◦ N - 90◦ N and 30◦ S - 90◦ S.

*Response: We added the following Fig. 3 in the future revised manuscript to show a direct model output of the basic experiment 1 and 2 with CE-NH and CE-SH:*

[Figure]

*Figure 3. The surface concentration (mol/mol) of the passive tracer averaged in January at each year of the simulation from 2015 to 2019. The subplots in the left column (a), (c), (e), (g), (i) show the passive tracer released from the NH in Experiment 1 and subplots in the right column (b), (d), (f), (h), (j) show the passive tracer released from the SH in Experiment 2. The blue lines and the red lines show the CE-NH and CE-SH respectively.*

*And we added the following sentences in the future revised manuscript to describe the distribution of the passive tracer: "The global distributions of the passive tracer averaged in January at each year of the simulation time from 2015 to 2019 are shown in Fig. 3. The concentration of the passive tracer*

*gradually increases after the releasing time in both experiment cases, where in the E1 the passive tracer is released in the NH and in the E2 the passive tracer is released in the SH. This latitudinal gradient can be clearly seen in the distribution of the passive tracer and is well determined by the CE-NH and CE-SH. The continuous release of the artificial tracer has been inspired by $SF_6$, which is also continuously released in the NH."*

Minor and technical comments:

- P1 L12

"......meaning the speed of the migration of the CE decreases with the altitude", do you mean the seasonality of the migration of the CE in the lower altitude is larger than that in the higher altitude?

*Response: yes, and we corrected the sentences as:*

*"We found that the vertical structure had a slight northern tilt in the NH winter season and a southern tilt in NH summer, meaning the seasonality of the migration of the CE in the lower altitude is larger than that in the higher altitude."*

- P2 L47

"......discussed. (Nicholson, 2009, 2018)." → "......discussed (Nicholson, 2009, 2018)."

*Response: corrected.*

- P3 L88

"......zonal range of 30◦ N - 90◦ N..." → "......zonal range of 30◦ N - 90◦ N and 30◦ S - 90◦ S..."

*Response: corrected.*

- P4 L92

"The time series of the tracer averaged zonally..." → "The time series of the tracer released from 30◦ N - 90◦ N averaged zonally..."

The figure presents two source domains. Please specify the source domain to avoid confusion.

*Response: the sentence was corrected. The following sentence was added to the revised manuscript:*

*"The source domains of the passive tracer are marked by shaded red and blue regions in 30° N - 90° N and 30° S - 90° S which means the passive tracer released from 30° N - 90° N (upper plot) and 30° S - 90° S (lower plot), respectively.*

*"*

[Figure]

*Figure 4. The releasing area of passive tracer (a) E1 (shown by shaded blue region in the upper plot) and (b) E2 (shown by shaded red region in the lower plot).*

• P4 L98-L99

"Since the variation around the trend does not vary with the level of the time series", what do you mean here? What is "the level of the time series"? Do you mean different time period or different vertical level? If it is different time period or altitude, I don't understand why the variation around the trend does not change.

*Response: we deleted these sentences since it is not necessary and introduced confusion.*

• P4 L103

"Figure 3 shows the decomposition of the time series of the passive tracer." → "Figure 3 shows the decomposition of the time series of the passive tracer released from 30◦ N - 90◦ N."

*Response: corrected.*

• P4 L112

"he longitude and attitude..." → "he longitude and latitude"

*Response: corrected.*

• P4 L112

"The trend in each grid box and each time step is meridionally averaged in the longitude of range of -180° to 180° and zonally averaged in the latitude range of 30° S to 30° N". It seems like this sentence might be" The trend in each time step is zonally averaged in the longitude of range of -180° to 180° and meridionally averaged in the latitude range of 30° S to 30° N"

*Response: we corrected the sentence as "The trend in each grid box and each time step is spatially averaged in the domain of -180° to 180° and 30° S to 30° N.*
*"*

• P4 L120-121

"For example, if the tracer concentration in a grid box is higher than Tt, this grid box is located on the NH, and vice versa for the SH." The authors should mention this is for the tracers originated from 30° N - 90° N.

*Response: corrected as: "For example, if the concentration of the tracer (released from 30° N - 90° N) in a grid box…*
*"*

• P5 L149

"Figure 4 shows the daily locations of the CE-SH and CE-SH..." → "Figure 4 shows the daily locations of the CE-NH and CE-SH..."

*Response: corrected.*

• P6 L156-L157

"The north of the boundary at 20° S is dominated by the airmass transport from NH". I'm not sure about this interpretation, especially for the region from 20° S to equator. Do you have the results about airmass distribution?

*Response: We removed this sentence to avoid confusion, and we have added Fig. 3 to show the distribution of the passive tracer from the model output.*

• P6 L157

"...except Africa..." → "...except Atlantic and Africa..."

*Response: corrected.*

• P6 L174-L177

It is better to introduce Figure 6 before Figure 7.

*Response: thanks, we have corrected it here and we reviewed the full text and corrected other similar figure labeling errors.*

• P6 L178

"...in the literature of (Fueglistaler et al., 2004)" → "...in the literature of Fueglistaler et al. (2004)"

*Response: corrected.*

• P6 L186-L87

"In the NH winter, … in boreal winter from December to February." repetitive winter in one sentence.

*Response: corrected to "In the NH winter from December to February, the CE-SH reaches its southernmost position at 15° S in South America."*

• P7 L203

"the CE-SH and CE-SH" → "the CE-NH and CE-SH".

*Response: corrected.*

• P7 L209-L210

"less than 2 km" → "below 2 km". "while the area above...", it is better to specify the latitude range.

"From April" → "In April and May"

*Response: we corrected here to "… air masses from the NH that are below 2 km move south of the geographic equator to about 10°S… In April and May …"*

• P7 L212

"while CE-SH slopes from the ground to 2 km and is relatively vertical to the ground", it might be better to change as "while CE-SH from the ground to 2 km is relatively vertical to the ground"

*Response: corrected.*

• P7 L217-L218

Like the question I addressed in the abstract, how to understand the movement speed of CE here? Is it monthly movement speed?

*Response: We corrected this sentence to:*

*"The seasonality of the migration of the CE in the lower altitude is larger than that in the higher altitude."*

• P8 L228

"the CE-SH / CE-SH..." → "the CE-SH / CE-NH".

*Response: corrected.*

• P10 L289-L290

"The north-south migration of the CE is consistent with the maximum rain rate during a year." It is not always consistent, especially in JJA (Figure 10).

*Response: corrected as "The north-south migration of the CE is not always consistent with the maximum rain rate during a year, especially in the TWP region."*

• P17

Figure 3(b): legend (6° S, 127.5° E)

Caption: (a) [6.0° N, 127.5° E] and (b) [6.0° S, 127.5° E].

*Response: corrected the legend and caption.*

• P22

It is better to include a vertical dashed line along latitude = 0 in Figure 8.

*Response: thanks, we have included a vertical line to show the latitude =0:*

[Figure]

*Figure 5. Monthly averaged (2015-2019) CE at the layers from the model vertical grids from surface to 8 km. The CE-SH / CE-NH are zonally (100° E-180°) averaged over the TWP region (30°S - 30°N, 100°E-180°). The blue lines show the CE-NH and the red lines show the CE-SH. The dashed black line shows the latitude =0. 1-σ of the CE-NH and CE-SH are given in the plots.*

- P23

Caption: This sentence "The data in this plot are also zonally averaged in the TWP region specified in Fig. 5." is not necessary since "averaged over the West Pacific region" was mentioned before in the caption.

*Response: corrected.*

- P24

Please include the coordinate in Figure 10.

*Response: Coordinators were added to the Figure.*

Recommendation

Read the manuscript thoroughly. Further improvements to the text clarity is necessary.

*Response: Thanks, we read the full text and made some revisions to make the article clearer and more accurate.*

*We add the following reference:*

*Veefkind, J. P., Aben, I., McMullan, K., Förster, H., Vries, J. d., Otter, G., Claas, J., Eskes, H. J., Haan, J. F. d., Kleipool, Q., Weele, M. v., Hasekamp, O., Hoogeveen, R., Landgraf, J., Snel, R., Tol, P., Ingmann, P., Voors, R., Kruizinga, B., Vink, R., Visser, H., and Levelt, P. F.: TROPOMI on the ESA Sentinel-5 Precursor: A GMES mission for global observations of the atmospheric composition for climate, air quality and ozone layer applications, Remote Sensing of Environment, 120, 70–83, https://doi.org/10.1016/j.rse.2011.09.027, 2012.*

*Schneising, O., Buchwitz, M., Reuter, M., Bovensmann, H., Burrows, J. P., Borsdorff, T., Deutscher, N. M., Feist, D. G., Griffith, D. W. T., Hase, F., Hermans, C., Iraci, L. T., Kivi, R., Landgraf, J., Morino, I., Notholt, J., Petri, C., Pollard, D. F., Roche, S., Shiomi, K., Strong, K., Sussmann, R., Velazco, V. A., Warneke, T., and Wunch, D.: A scientific algorithm to simultaneously retrieve carbon monoxide and methane from TROPOMI onboard Sentinel-5 Precursor, Atmospheric Measurement Techniques, 12, 6771–6802, https://doi.org/10.5194/amt-12-6771-2019, 2019.*

*Schneising, O., Buchwitz, M., Hachmeister, J., Vanselow, S., Reuter, M., Buschmann, M., Bovensmann, H., and Burrows, J. P.: Advances in retrieving XCH4 and XCO from Sentinel-5 Precursor: improvements in the scientific TROPOMI/WFMD algorithm, Atmospheric Measurement Techniques, 16, 669–694, https://doi.org/10.5194/amt-16-669-2023, publisher: Copernicus GmbH, 2023.*

*Muntean, M., Janssens-Maenhout, G., Song, S., Giang, A., Selin, N. E., Zhong, H., Zhao, Y., Olivier, J. G., Guizzardi, D., Crippa, M., Schaaf, E., and Dentener, F.: Evaluating EDGARv4.tox2 speciated mercury emissions ex-post scenarios and their impacts on modelled global and regional wet deposition patterns, Atmospheric Environment, 184, 56–68, https://doi.org/https://doi.org/10.1016/j.atmosenv.2018.04.017, 2018.*

---

## Author Comment (AC3)

Response to Comments of Reviewer 2

*The authors thank all reviewers for their constructive comments and suggestions, which have helped us improve this paper's quality in sciences and writing. All comments are carefully considered and responded to. The response in blue italic letters follows each comment in black.*

The paper, "Determination of the Chemical Equator from GEOS-Chem Model Simulation: A Focus on the Tropical Western Pacific Region" by Xiaoyu Sun et al., proposes a method to identify the hemispheric boundary of air mass transport known as the chemical equator, which has not been extensively studied. The study concentrates on the Tropical West Pacific region and utilizes artificial tracers simulated by the GEOS-Chem model to determine the chemical equator's location. The authors investigate the chemical equator's vertical structure and compare it to the tropical rain belt and wind field convergence. The article fits within the scope of ACP, but significant revisions are necessary before it can be published. To improve the manuscript, the following revisions are suggested. Clarify the vertical level of the chemical equator's results to better understand the location of air mass transport between the hemispheres. Strengthen the justification for the chemical equator determination method by providing a more detailed explanation of the reasoning behind the methodology used. Provide additional evidence to support the interpretation of some results. This can be achieved through the inclusion of further analysis or data that reinforces the conclusions drawn from the results. Apply the chemical equator concept to gain a better understanding of inter-hemispheric air transport. This could include investigating how the chemical equator affects atmospheric composition or analyzing how it may influence global climate patterns.

The following revisions are proposed:

1. To better understand the inter-hemispheric transport of pollutants/tracers, the manuscript should investigate the relationship between the chemical equator's location and atmospheric compositions such as CO, $CH_4$, and $SF_6$. This analysis could utilize satellite observations or model output to provide a comprehensive understanding of the differences in tracer distribution between the northern and southern hemispheres. Including a TransCom simulation analysis (i.e. Krol et al., GMD, 2018) in the discussion would be a valuable addition to the manuscript.

*Response: Thanks for the suggestion. We added the following discussion to the future revised manuscript in the Sect. 1:*

*"Methane ($CH_4$) and Carbon dioxide ($CO_2$) have a relatively long lifetime and clear latitudinal gradient which has the potential as tracers to investigate the interhemispheric transport. The model output such as TransCom of $CO_2$ provide a comprehensive understanding of the differences in tracer distribution between the northern and southern hemispheres and study the IHT (Krol et al., GMD, 2018)."*

*We added the comparison of the CE with the global distribution of vertical columns of $CH_4$ from TROPOMI and the surface concentration of $SF_6$ from the GEOS-Chem simulation.*

*We added Sect. 3.2 to describe the connection between the CE and the distribution of $CH_4$ and $SF_6$:*

*"Section 3.2 The Chemical Equator and the distribution of atmospheric compositions*

*To better understand the implication of the CE position, satellite measurements of $CH_4$ and model simulation of $SF_6$ are presented together with the CE in Fig. 2 . $CH_4$ used in this study is retrieved from TROPOMI measurements aboard in Sentinel-5 Precursor satellite mission (Veefkind et al., 2012) in the SWIR wavelengths (2300-2389 nm). Here we use the latest release of the WFMD (Weighting Function Modified Differential Optical Absorption Spectroscopy) product (v1.8) (Schneising et al., 2023) and*

*process it onto a 5° x 5° grid. The details of the satellite data product is described in the Appendix C. We used GEOS-Chem v13.0.0 to obtain the simulation of $SF_6$. The model set-up of $SF_6$ is described in details in Appendix D. The CE and the global distribution of $CH_4$ and $SF_6$ averaged for January and July 2019 are shown in Fig. 2. The CE and the north-south gradient of $CH_4$ in the Indian Ocean in January 2019 are well consistent with each other. This indicates the CE has good potential to illustrate the IHT inferred by the satellite measurements of $CH_4$. However, due to the lack of data coverage, it is relatively difficult to see the distribution of $CH_4$ in SH from the satellite measurement in July. The $CH_4$ distribution is also affected by 1) sources in the SH and b) removal due to OH. This means the $CH_4$ concentration is not monotonically rising like the inert artificial tracer used in our study and does not show a clear distinction between NH and SH. $SF_6$ has this property and has been used for similar purposes (e.g., Geller et al., 1997; Waugh et al., 2013; Yang et al., 2019), but there are large emissions in South East Asia, which may be emitted into the CE area."*

*And we added Appendix to describe the satellite products of $CH_4$ we use and the GEOS-Chem model set-up of the $SF_6$.*

*"Appendix C*

*The Sentinel-5 Precursor satellite mission (Veefkind et al., 2012) was launched on 13 October 2017 carrying a single scientific instrument, TROPOMI, which is a nadir viewing passive grating imaging spectrometer. The satellite is positioned in a near-polar, sun-synchronous orbit and has a swath width of 2600 km, allowing for daily Earth coverage. The retrieval is however dependent on sun-lit, cloud-free scenes which limits the daily coverage. The instrument consists of four spectrometers measuring radiances in the ultraviolet, ultraviolet-visible, near-infrared, and short-wave infrared bands. $CH_4$ used in this study is retrieved from TROPOMI measurements of sunlight reflected by Earth's surface and the atmosphere in the SWIR wavelengths (2300-2389 nm). The spatial resolution is 5.5x7 $km^2$. The Weighting Function Modified Differential Optical Absorption Spectroscopy (WFMD) TROPOMI data product (Schneising et al., 2019) provides vertical columns of both methane $CH_4$ and CO. Here we use the latest release of the WFMD product (v1.8) (Schneising et al., 2023) and process it onto a 5° x 5° grid. For this, each measurement is assigned to a single grid cell and the weighted average of all measurements per cell is calculated. The measurements are weighted using the inverse standard deviation to disadvantage measurements with high uncertainty. Additionally, only measurements with a quality flag (qf) qf=0 (good) are included. Data coverage is therefore limited over regions with many clouds (e.g. tropics) or challenging measurement conditions.*

*Appendix D*

*The meteorological fields used in the model are from MERRA-2 reanalysis as described in Sec. 2.1. We performed the simulation of $SF_6$ from 2014 to 2019 in the horizontal grid resolution of 2° x 2.5° and vertical grid resolution of 72 levels. The emission database of $SF_6$ is annually spatially- girded and taken from the Emission Database for Global Atmospheric Research (EDGAR version 4.2) inventory (Muntean et al., 2018), available at 0.1° x 0.1° global resolution for 1970-2008.*

*"*

*And we added the following Figure 1 in the future revised manuscript:*

[Figure]

*Figure 1. CE with Sentinel-5 Precursor satellite CH₄ vertical columns (ppbv) averaged for (a) January 2019 and (b) July 2019. CE with SF₆ surface concentration (ppbv) simulated by GEOS-Chem averaged for (c) January 2019 and (d) July 2019. The blue dots show the NH boundary (CE-NH) and the red dots show the SH boundary (CE-SH).*

2. The manuscript should discuss the unique features of the chemical equator over the Tropical Western Pacific region compared to other regions. This could include a detailed analysis of the chemical equator's behavior and characteristics in the context of the Western Pacific Monsoon and other regional circulations.

*Response: The CE is a tool to determine the boundary for air mass transport on a global scale. So the boundary determined by the CE basically has no unique feature in the TWP region compared to other regions over the tropics, despite the complicated circulation pattern. This is the reason why the use of the ITCZ fails to clearly separate the hemispheres in the TWP. But our study and others using models (Hamilton et al., 2008) show that such a separation exists.*

*However, the reason we choose to focus on the TWP region is that this region is considered as the major transport pathway from the troposphere into the stratosphere during the NH winter. So the air mass transport and the origins in this region need to be studied in detail, and the main application of the CE in this study is to investigate the air mass transport in the TWP. But the method of the CE can also be used for similar studies in other parts of the world, i.e. Africa or South America, which also show a complicated circulation due to the orography.*

3. To better understand air mass transport, the manuscript should include the distribution of artificial tracers from 30° N - 90° N and 30° S - 90° S. This will provide insight into the relative contributions from source domains and air mass inter-hemispheric transport.

*Response: We added the following Fig. 2 in the future revised manuscript to show a direct model output of the basic experiment 1 and 2 with CE-NH and CE-SH:*

[Figure]

*Figure 2. The surface concentration (mol/mol) of the passive tracer averaged in January at each year of the simulation from 2015 to 2019. The subplots in the left column (a), (c), (e), (g), (i) show the passive tracer released from the NH in Experiment 1 and subplots in the right column (b), (d), (f), (h), (j) show the passive tracer released from the SH in Experiment 2. The blue lines and the red lines show the CE-NH and CE-SH respectively.*

*And we added the following sentences in the future revised manuscript to describe the distribution of the passive tracer: "The global distribution of the passive tracer averaged in January at each year of the simulation time from 2015 to 2019 are shown in Fig. 2. The concentration of the passive tracer gradually increases after the releasing time in both experiment cases, where in the E1 the passive tracer is released in the NH and in the E2 the passive tracer is released in the SH. This latitudinal gradient can be clearly seen in the distribution of the passive tracer and is well determined by the CE-NH and CE-SH. The continuous release of the artificial tracer has been inspired by $SF_6$, which is also continuously released in the NH."*

4. The manuscript should clarify the vertical level on which the results are based to obtain the chemical equator. This information should be provided in both the methods and results sections to ensure that the reader can understand the study's findings.

*Response: The results in the paper are based on the vertical grids of GEOS-Chem which are 72 layers from the surface up to 10 hPa / 80 km. We checked the whole manuscript and made the following changes:*

- *We added the sentence:*

*"The simulation results used in this study are based on the vertical and horizontal grids which are 72 levels and 0.5°x0.625°, respectively."*

- *And we corrected the sentence:*

*"We calculate CE at each vertical grid of the model output of the passive tracer. When the level goes up, it becomes hard to find an actual boundary between the two hemispheres due to the fast horizontal mixing by high-speed winds in the upper troposphere and lower stratosphere. So, we only take the level of the model grids under 8 km into consideration."*

- *And we corrected the caption of Figure 9 in the original manuscript like this:*

*"Figure 9. Monthly averaged (2015-2019) CE at the layers from the model vertical grids from surface to 8 km. The CE-NH / CE-SH are zonally (100° E-180°) averaged over the TWP region see Fig. 6. The blue lines show the CE-NH and the red lines show the CE-SH. 1-σ of the CE-NH and CE-SH are given in the plots. "*

5. To improve the manuscript's clarity, it is recommended to conduct a thorough review to identify inaccuracies, misprints, and errors. These revisions will enhance the text's readability and improve the manuscript's overall quality.

*Response: Thank you. We read the entire text and made some edits to improve the article's accuracy and clarity.*

*We add the following reference:*

*Krol, M., de Bruine, M., Killaars, L., Ouwersloot, H., Pozzer, A., Yin, Y., Chevallier, F., Bousquet, P., Patra, P., Belikov, D., Maksyutov, S., Dhomse, S., Feng, W., and Chipperfield, M. P.: Age of air as a diagnostic for transport timescales in global models, Geoscientific Model Development, 11, 3109–3130, https://doi.org/10.5194/gmd-11-3109-2018, 2018.*

*Veefkind, J. P., Aben, I., McMullan, K., Förster, H., Vries, J. d., Otter, G., Claas, J., Eskes, H. J., Haan, J. F. d., Kleipool, Q., Weele, M. v., Hasekamp, O., Hoogeveen, R., Landgraf, J., Snel, R., Tol, P., Ingmann, P., Voors, R., Kruizinga, B., Vink, R., Visser, H., and Levelt, P. F.: TROPOMI on the ESA Sentinel-5 Precursor: A GMES mission for global observations of the atmospheric composition for climate, air quality and ozone layer applications, Remote Sensing of Environment, 120, 70–83, https://doi.org/10.1016/j.rse.2011.09.027, 2012.*

*Schneising, O., Buchwitz, M., Reuter, M., Bovensmann, H., Burrows, J. P., Borsdorff, T., Deutscher, N. M., Feist, D. G., Griffith, D. W. T., Hase, F., Hermans, C., Iraci, L. T., Kivi, R., Landgraf, J., Morino, I., Notholt, J., Petri, C., Pollard, D. F., Roche, S., Shiomi, K., Strong, K., Sussmann, R., Velazco, V. A., Warneke, T., and Wunch, D.: A scientific algorithm to simultaneously retrieve carbon monoxide and methane from TROPOMI onboard Sentinel-5 Precursor, Atmospheric Measurement Techniques, 12, 6771–6802, https://doi.org/10.5194/amt-12-6771-2019, 2019.*

*Schneising, O., Buchwitz, M., Hachmeister, J., Vanselow, S., Reuter, M., Buschmann, M., Bovensmann, H., and Burrows, J. P.: Advances in retrieving XCH4 and XCO from Sentinel-5 Precursor: improvements in the scientific TROPOMI/WFMD algorithm, Atmospheric Measurement Techniques, 16, 669–694, https://doi.org/10.5194/amt-16-669-2023, publisher: Copernicus GmbH, 2023.*